# Use of Surface Electromyography to Estimate End-Point Force in Redundant Systems: Comparison between Linear Approaches

**DOI:** 10.3390/bioengineering10020234

**Published:** 2023-02-10

**Authors:** Daniele Borzelli, Sergio Gurgone, Paolo De Pasquale, Nicola Lotti, Andrea d’Avella, Laura Gastaldi

**Affiliations:** 1Department of Biomedical and Dental Sciences and Morphofunctional Imaging, University of Messina, 98124 Messina, Italy; 2Laboratory of Neuromotor Physiology, IRCCS Fondazione Santa Lucia, 00179 Rome, Italy; 3Center for Information and Neural Networks (CiNet), Advanced ICT Research Institute, National Institute of Information and Communications Technology, Suita City 565-0871, Osaka, Japan; 4Institut fur Technische Informatik (ZITI), Heidelberg University, 69120 Heidelberg, Germany; 5Department of Mechanical and Aerospace Engineering, Politecnico di Torino, 10129 Turin, Italy

**Keywords:** myoelectric control, EMG-to-force mapping, linear regression, ridge regression, opensim, musculoskeletal model, muscle synergies, co-contraction, longitudinal study, muscle stiffnes

## Abstract

Estimation of the force exerted by muscles from their electromyographic (EMG) activity may be useful to control robotic devices. Approximating end-point forces as a linear combination of the activities of multiple muscles acting on a limb may lead to an inaccurate estimation because of the dependency between the EMG signals, i.e., multi-collinearity. This study compared the EMG-to-force mapping estimation performed with standard multiple linear regression and with three other algorithms designed to reduce different sources of the detrimental effects of multi-collinearity: Ridge Regression, which performs an L2 regularization through a penalty term; linear regression with constraints from foreknown anatomical boundaries, derived from a musculoskeletal model; linear regression of a reduced number of muscular degrees of freedom through the identification of muscle synergies. Two datasets, both collected during the exertion of submaximal isometric forces along multiple directions with the upper limb, were exploited. One included data collected across five sessions and the other during the simultaneous exertion of force and generation of different levels of co-contraction. The accuracy and consistency of the EMG-to-force mappings were assessed to determine the strengths and drawbacks of each algorithm. When applied to multiple sessions, Ridge Regression achieved higher accuracy (R^2^ = 0.70) but estimations based on muscle synergies were more consistent (differences between the pulling vectors of mappings extracted from different sessions: 67%). In contrast, the implementation of anatomical constraints was the best solution, both in terms of consistency (R^2^ = 0.64) and accuracy (74%), in the case of different co-contraction conditions. These results may be used for the selection of the mapping between EMG and force to be implemented in myoelectrically controlled robotic devices.

## 1. Introduction

A measure of muscle activation is provided by electromyography (EMG), i.e., the recordings of electrical activity in muscle fibers driven by motoneurons. While the EMG signal recorded through needle electrodes provides an accurate measure of a small volume of the muscle, applications that require the modulation of the whole muscle commonly require the use of non-invasive surface EMG. Surface EMG has been implemented in industrial applications, or research studies on motor control, confining the use of needle EMG to clinical applications, or to research investigating the recruitment of single motor neurons [1]. The activation of muscles has also been exploited to estimate the end-point force generated by a human operator. In the last decades, several myoelectrically controlled robotic devices, such as prostheses [2] and exoskeletons [3], have been developed. In fact, the EMG signal allows to noninvasively convey information about the operator’s intent [4,5,6,7], and to myoelectrically control a robotic device [8]. The simplest myoelectric control scheme employs a few pairs of antagonist muscles, each controlling a single degree of freedom [4]. However, these systems are inherently limited, since each muscle might act on multiple Degrees of Freedom (DoFs) [9] and may be sensitive to noise [10].

An accurate estimation of end-point force through the EMG signals of multiple muscles can be achieved by EMG-driven musculoskeletal models [11,12]. However, their use requires long calibration, implementation of specific models for the task [13], and subject-specific models [14,15]. Therefore, during isometric tasks, in which the non-linearities in the dependence of muscle tension, muscle contraction velocity, and muscle length are not present, the relation between EMG activity and end-point force may be approximated by linear mapping [16,17,18,19].

A simple approach that estimates the EMG-to-force linear mapping involves multiple linear regression analysis, which minimizes the reconstruction error of the exerted end-point forces by a linear combination of the EMG signal [17,19,20,21]. However, the multiple linear regression approach may result in an inaccurate estimation due to the noise of the collected EMG signals [22] and multicollinearity. Multicollinearity occurs when some explanatory variables, which in this scenario are the muscle activations, are not independent and lead to the fitting of noise [23]. A correlation between muscle activations could be a consequence of crosstalk [24], or it could derive from physiological factors such as the synergistic recruitment, possibly through common synaptic inputs, of different muscles. According to the muscle synergy hypothesis [25,26,27], groups of muscles are recruited with specific and task-independent activation balances (spatial synergies) or waveforms (temporal synergies) to generate all required end-point forces by task-dependent synergy activation coefficients. Therefore, a few synergistic synaptic inputs would drive multiple muscles, reducing the dimensionality of the control signals, but also leading to a physiological correlation in the muscle activities. Similarly, several studies demonstrated the existence of a common input to multiple synergistic [28,29,30] or non-synergistic [31,32,33,34] muscles. This common input synchronizes the firings of pools of motor neurons of different muscles and, therefore, the activity of these muscles. Both muscle synergies and common input would represent a physiological source of multicollinearity, which cannot be neglected as it would lead to the reduction in the quality of the reconstruction of exerted end-point forces from multiple EMG signals.

Different approaches have been proposed to overcome the detrimental effect of multicollinearity in estimating linear mappings [35]. The Ridge Regression (RR) algorithm adds a weight parameter that modifies the explanatory variables to reduce the linear relation between the variables [36]. Another approach consists of implementing foreknown boundaries in the regression [37,38]. A third approach consists in reducing the number of degrees of freedom of the system to a set of independent variables [35]. 

The aim of this study was to compare different approaches for the estimation of the EMG-to-force mapping based on linear regression and other approaches which specifically target the reduction of the effects due to different sources of multicollinearity. This comparison may be useful in the selection of the control logic of myoelectrically controlled robotic devices driven by the activities of multiple muscles and may provide an approach to indirectly infer the main source of the dependence between the activities of different muscles in a dataset. The metrics used for the comparison are the quality and consistency of data fitting and computational cost. The approaches included in the comparison are: (1) standard multiple linear regression (LR), (2) RR, (3) linear regression with foreknown anatomical constraints (AC) derived through a validated musculoskeletal model, i.e., Opensim [39], and (4) multiple regression on a reduced set of the degrees of freedom of the musculoskeletal system provided by the activation coefficients of muscle synergies (MS).

The consistency in the estimation of the EMG-to-force mapping was validated by testing the selected approaches on data collected during two isometric force generation protocols in different experimental sessions and different conditions. In the first protocol, participants were asked to exert the same set of isometric end-point forces in five different sessions, spanning two days. In the second protocol, participants were asked to exert the same set of isometric end-point forces in two conditions: with and without a voluntary increase of muscular co-contraction for limb stiffness modulation. 

The present work is organized as follows. In Section 2, the experimental setups and protocols are presented, and the algorithms for the identification of the EMG-to-force mappings and the analyses implemented to compare them are described. The findings are reported in Section 3. Finally, the Discussion and Conclusions are found in Section 4 and Section 5, respectively.

## 2. Materials and Methods

In this study, we used data previously presented in two different published studies: protocol 1 with five different sessions in [40] and protocol 2 with different levels of co-contraction in [41]. The setup and the protocols are briefly described below.

### 2.1. Participants

Five right-handed subjects (mean (SD); age 25.6(4.7) years; body mass: 74(10) kg; height: 1.77(0.06) m; 1 female) participated in protocol 1 and nine right-handed subjects (age 23.8(3.5) years; body mass: 71(4) kg; height: 1.72(0.06) m; 6 females) participated in protocol 2 after giving written informed consent. All procedures were conducted in conformity with the Declaration of Helsinki and were approved by the Ethical Review Board of IRCCS Neurolesi ‘Bonino Pulejo’ (protocol 1) or the Ethical Review Board of the IRCCS Fondazione Santa Lucia (protocol 2).

### 2.2. Setup and Protocols

During both protocols, participants inserted their hand and forearm into an orthosis that was rigidly connected to a 6-axis force and torque transducer (Figure 1A). Their hand was pronated while the elbow was flexed at 90°. To ensure isometric conditions, participants’ trunks and shoulders were strapped to a chair through car safety belts. Participants were asked to exert forces with their hand along different directions in the space. They were provided with feedback, displayed by a 3D monitor, as the displacement of a virtual spherical cursor whose position was proportional to the exerted force. Participants wore 3D glasses to perceive the scene in depth through stereoscopic vision. The cursor motion was simulated as a mass-spring-damper system moved by force exerted by the participant (MSD1 in Figure 1E). In protocol 1, pictures of the electrodes’ placement were taken at the beginning of the experiment for precise replacing in the following sessions.

For both protocols, all trials (Figure 1B) started with a rest phase, in which the participant had to maintain the cursor within a spherical target centered in the zero position and with a tolerance radius of 2% of the Maximum Voluntary Force (MVF). During the following dynamic phase, the participant had to move the cursor to a displayed force target, and then keep it within the target 2% tolerance (static phase). 

#### 2.2.1. Protocol 1

The experiment lasted 2 days and was composed of 5 sessions. During day 1, sessions 1 to 4 were performed, while the last session was performed during day 2 (see Figure 1C). During each session, the MVF, exerted along 8 equally spaced directions on the horizontal plane, was calculated. Then the participant was asked to move the cursor into targets which were 10%, 20%, and 30% of the mean MVF isometric forces from the rest position, and along 8 equally spaced planar target directions, 3 repetitions each, and to keep the virtual cursor within a 2% MVF threshold for 2 s. Between the first and the second session, a small change in posture was induced by asking the participant to extract the forearm from the orthosis, while a higher postural variation was induced between the second and the third sessions by asking the participant to stand up. Between the third and the fourth session, an accidental detachment of EMG electrodes was simulated, and all electrodes were removed and repositioned, according to the small marks on the skin that were left by the electrodes. The last session (session 5) was performed on a different day, during which electrode positioning was guided by the photographs taken in session 1. 

#### 2.2.2. Protocol 2

The experiment was subdivided into 6 blocks (Figure 1D) all performed on the same day. During the initial MVF block, participants generated forces along 20 directions, that were aligned with the vertices of a dodecahedron centered in the rest position. During the next baseline block of force control (FC), participants displaced the virtual cursor to one of the 20 targets, positioned at the vertices of a dodecahedron inscribed into a sphere with a 20% MVF radius and centered in the rest position. Participants were instructed to maintain the cursor within the spherical target, with a 3% MVF tolerance, for 1 s. During the third block of pure co-contraction (CC), participants practiced reducing the cursor oscillation by co-contracting their muscles to familiarize themselves with the co-contraction task. However, data collected during this block were not analyzed in this study. The last three perturbed (P1–P3) blocks, were similar to FC, i.e., participants had to maintain the cursor within one of the 20 force targets for 1 s, but during these blocks, the cursor motion around the mean position was modeled as a mass-spring-damper system (MSD2 in Figure 1E) perturbed by a sinusoidal force whose amplitude increased along the perturbed blocks. The effects of the external deflective force were reduced by participants by increasing the stiffness of MSD2 (‘virtual stiffness’ [41]). This was simulated as a function of the norm of the instantaneous projection of the muscle activation onto the null space component of the EMG-to-force mapping. The EMG-to-force mapping was calculated online as the linear regression between EMG and force. The relation between the null space component of the muscle activation was demonstrated to be related to limb stiffness [42,43]. The FC and the P1-P3 blocks were composed of 60 trials each, as three repetitions for each of the 20 targets were presented.

### 2.3. EMG Acquisition

Bipolar EMG signals were recorded with surface active bipolar electrodes (Trigno wireless, Delsys Inc., Natick, MA, USA, during protocol 1 and Bagnoli-16, Delsys Inc., Natick, MA, USA, during protocol 2) from 15 muscles during protocol 1 and 17 muscles during protocol 2. However, muscles that were not collected in both protocols or that were not modeled by the upper-limb Opensim model [44] used to define the anatomical constraints (see below) were excluded. Therefore, only the activities of 12 muscles were investigated in this study, which were: (1) brachioradialis, (2) biceps brachii long and (3) short heads, (4) pectoralis major, (5) anterior deltoid, (6) middle deltoid, (7) posterior deltoid, (8) triceps brachii lateral head and (9) long head, (10) infraspinatus, (11) teres major, and (12) latissimus dorsi. 

EMG activity was acquired at 1000 Hz, bandpass filtered (20–450 Hz), amplified with a 1000 gain. Electrodes were positioned on the subjects’ skin, which was cleansed with alcohol, according to the recommendations proposed by SENIAM [45] and by palpating muscles to locate the muscle belly. The electrodes were oriented along the main fiber direction to maximize the signal-to-noise ratio [46]. Force and EMG data were digitized at 1 kHz using an analog-to-digital PCI board (PCI-6229; National Instruments, Austin, TX, USA). Data acquisition and analysis were performed with custom software written in Matlab^®^ (MathWorks Inc., Natick, MA, USA) and Java.

### 2.4. EMG-to-Force Matrix Estimations

The EMG-to-force matrix ***H***, whose dimensions were (force dimensions × number of muscles) therefore (3 × 12) in this study, approximated with a linear relation [47,48] the mapping of the EMG signal ***m***, whose dimensions were (number of muscles × number of samples) onto the end-point force ***f***, whose dimensions were (force dimensions × number of samples):(1)f=H m

EMG data were baseline subtracted, rectified, 2nd order Butterworth filtered with 1 Hz cutoff frequency, baseline subtracted again, normalized to the peak collected during the MVF block, and resampled at 100 Hz. The force data were 2nd order Butterworth filtered at 1 Hz cutoff and resampled at 100 Hz.

EMG-to-force matrices were estimated from experimental EMG and force data. Different matrices were calculated from data collected during each session of protocol 1 and collected during baseline block and grouped perturbed blocks in protocol 2, with each of the selected algorithms, as described in the following.

#### 2.4.1. Unconstrained Standard Linear Regression (LR) Approach

In the unconstrained standard LR approach, the EMG-to-force matrix was calculated as the linear regression of the EMG signals on each end-point force component. The LR approach identified the mapping which allowed to estimate forces from the experimental EMG data best fitting the recorded forces. Given a vector ***f_i_***, indicating the samples of the *i*-th component of the exerted end-point forces, whose dimensions are (1 × *n*) (i.e., a row vector, *n* is the number of time samples), and a matrix ***M***, indicating the EMG data collected from *m* muscles, whose dimensions are (m × n), the linear regression estimates the coefficients $\beta_{i}$, whose dimensions are (1 × m), such that:(2)$$f_{i}=\beta_{i}\;M$$

The LR approach relies on least-square estimation, where the coefficients are estimated as:(3)$${\hat{\beta }}_{i\_LR}={\left(M^{T}M\right)}^{-1}M^{T}f_{i}$$
where ${\hat{\beta }}_{i\_LR}$ are the estimated coefficients with the LR approach and $M^{T}$ indicates the transposed EMG data matrix. In this study, the LR approach was performed with the Matlab function regress.

#### 2.4.2. Unconstrained Ridge Regression (RR) Approach

RR [36,49,50], or L2 regularization, is an algorithm that performs a linear regression on data, and it is particularly efficient for data suffering from multicollinearity [51,52]. In fact, when terms are correlated, the matrix ${\left(M^{T}M\right)}^{-1}$ in Equation (3) is close to singular and the ${\hat{\beta }}_{i\_LR}$ estimation would be highly affected by errors in the acquired data. Ridge regression solves the multicollinearity issue by estimating regression coefficients as follows:(4)$${\hat{\beta }}_{i\_RR}={\left(M^{T}M+k_{i}I\right)}^{-1}M^{T}f_{i}$$
where ***I*** is the identity matrix and the penalty term ***k_i_*** is called the “ridge parameter”, one for each force component. However, the parameters ***k_i_*** are not defined a priori. The ***k_i_*** parameters that best fit the data reducing the effect of multicollinearity are identified after determining the minimum of the mean square error as a function of the ridge parameter [36]. In this study, we calculated the mean square error at different ***k_i_*** that ranged from 0 (i.e., linear regression) and increased with steps of 0.001 until a minimum was identified. The RR approach was performed with the Matlab function ridge and the predictors were centered and scaled to have zero mean and standard deviation one.

#### 2.4.3. Constrained to Anatomical Musculoskeletal Foreknow Boundaries (Anatomically Constrained, AC) Approach

The algorithm for the estimation of the EMG-to-force matrix through the AC approach consisted of two steps: first, the Biomechanical ToolKit (b-tk) [53], driven by Matlab, calculated an initial mapping on a scaled upper-limb musculoskeletal model, which implemented anatomically accurate boundaries. This initial estimation was then implemented as anatomical constraints for a second mapping, which optimized the experimental data. The algorithm is described below.

Setting the posture of the model: The joint angles of the OpenSim MoBL-ARMS model [44] were set accordingly to the posture that all participants approximately assumed during the experiment. The angle of the joints, defined according to the reference system implemented in the Opensim Model, were: Wrist Flexion angle: 0°; Wrist deviation angle: 0°; Wrist prono-supination angle: 30°; Elbow flexion angle: 90°; Shoulder rotation angle: 60°; Shoulder abduction-adduction angle: 65°; Shoulder flexion-extension angle: 55° (Figure 2).

Model scaling: The mass of the model, the length of each segment, and the maximum isometric force of each muscle, were scaled to generate an upper limb subject-specific model. The scaling was performed by the Opensim Scaling tool by matching participants’ anthropometry, considering the height and the body mass of the participant [54].

The mass of the scaled model ($m_{m_{s}}$) was determined by the following equation:(5)$$m_{m_{s}}=m_{s}\;\;m_{m_{UL}}/m_{m_{FB}}$$
where $m_{s}$ was the mass of the participant, $m_{m_{UL}}$ the mass of the MoBL-ARMS model (i.e., 4.7782 kg), and $m_{m_{FB}}$ the full body model mass (i.e., 75.16 kg [55]).

The length of the segments of the model was uniformly scaled in three dimensions by the factor $S_{l}$:(6)$$S_{l}=d_{s}/d_{m}$$
where $d_{m}$ was the shoulder-elbow distance of the MoBL-ARMS model (i.e., 0.2837 m) and $d_{s}$ was the corresponding length measured on the participant.

Scaling of the maximum isometric force: The maximum isometric force that each muscle may exert was scaled by a factor ($S_{f}$) defined as in [54]:(7)$$S_{f}=\frac{h_{s}\;m_{s}}{h_{m_{FB}}m_{m_{FB}}}$$
where $h_{\mathrm{FB}}$ was the height of the 50% male percentile (1.7 m [55]) and $h_{s}$ the height of the participant. Therefore, the maximum isometric force of each muscle *i* ($F_{MAX_{i}}$) was:(8)$$F_{MAX_{i}}=S_{f}\;F_{MAX_{i}}^{m}$$
where $F_{MAX_{i}}^{m}$ was the maximum isometric force exerted by the muscle *i*, as implemented in the non-scaled model.

*Calculation of the moment arms*: The OpenSim Inverse Kinematic tool was used to estimate the moment arm of each muscle, in the pose assumed by the model, regarding each joint.

*Calculation of the inverse Jacobian*: The OpenSim Inverse Dynamic tool was exploited to calculate the inverse Jacobian. Seven simulated forces (0 N, ±0.1 N, ±0.2 N, and ±0.3 N) were applied at the model endpoint (i.e., the hand) along each of the x, y, and z axes, and the corresponding torque was calculated at each joint. Therefore, a linear regression determined the slope between the simulated forces along the *i*-th axis, and the corresponding torques calculated by the Inverse Dynamic tool, which was assigned as the component of the inverse Jacobian along the *i*-th axis.

*Calculation of the non-optimized constrained EMG-to-force matrix*: An EMG-to-force matrix, which was still non-optimized on the experimental data, was estimated as follows:(9)$$H_{non-optimized}=J^{-1}M\;F_{MAX}$$
where $F_{MAX}$ was a diagonal matrix, whose elements are the scaled maximum isometric force of each muscle, that mapped muscle activation onto muscle force; ***M*** the matrix of moment arms that mapped muscle forces onto joint torques; and $J^{-1}$ the inverse Jacobian matrix that mapped the joint torques onto the end-point force.

*Optimization of the constrained EMG-to-force matrix*: as the algorithm for the estimation of the $H_{non-optimized}$ matrix did not take the experimental data into consideration, it led to a poor fit of the endpoint force from EMG signals, despite its anatomical accuracy. Thus, another EMG-to-force mapping ($H_{AC}$) was built by constraining the muscle-pulling vectors (i.e., the matrix columns) to assume a direction that does not deviate more than 45° from the one of $H_{non-optimized}$, while the ratio between the amplitude of the $H_{non-optimized}$ and the $H_{constrained}$ pulling vectors was assumed to be in the range [0.5 2]. The $H_{AC}$ mapping was calculated through a global optimization of the force data with EMG data (Matlab function GlobalSearch)

#### 2.4.4. Degree of Freedom Reduction According to Muscle Synergies (MS) Approach

The muscle synergy hypothesis poses that muscles are recruited in fixed groups (i.e., muscle synergies) that are independent of the task. The existence of muscle synergies has been supported by the observation of low dimensionality in the muscle patterns during several tasks, including reaching and grasping [25,26], postural control [56,57], locomotion [58,59,60], and isomeric force generation [21,61], while the neural origin of muscle synergies has been supported through different approaches [16,20,62,63,64,65].

As muscle synergies represent the primitive modules driving the motor control and identify a limited number of DoFs with respect to the muscles, they are expected to be independently controlled by the Central Nervous System. Therefore, an estimation of the EMG-to-force mapping accounting for muscle synergies would represent a set of independent variables. A similar approach would be to reduce the number of muscles to a smaller set of agonist/antagonist pairs [66,67,68,69]. However, agonist/antagonist muscles are not defined for multi-muscle and multi-joint tasks (e.g., for biarticular muscles).

For calculating the MS EMG-to-force mapping, we separately extracted the muscle synergies from the EMG data collected during each session of protocol 1 and the baseline and the perturbed blocks of protocol 2, through the non-negative matrix factorization (NMF) algorithm [70]. The NMF algorithm allows to decompose a multidimensional time-varying signal ***m***(*t*), such as the EMG, into a matrix of modules ***W***, i.e., the muscle synergies, and a vector of time-varying coefficients ***c****(t)*, as follows:(10)m(t)=W c(t)
where *t* is the time of each sample.

The number of synergies was set equal to *n*, with *n* = 1, …, *N*, where *N* = 12 was the number of muscles collected in the experiments, and 10 repetitions of the synergy extraction were performed for each *n*. The repetitions showing the best reconstruction of EMG data were retained. The number of synergies was selected as the smallest number that could reconstruct the data with a fraction of data variation explained (*R*^2^) equal to or greater than 90%.

For each dataset used to identify synergies, a synergy-to-force mapping $H_{syn-to-force}$ was identified through standard linear regression of the synergy coefficients on each force component.

Therefore, the EMG-to-force mapping $H_{MS}$ was estimated as follows:(11)$$H_{MS}=H_{syn-to-force}\;W^{+}$$
where $W^{+}$ is the pseudo-inverse of the synergy matrix ***W***.

### 2.5. Statistics

The total computational time required to obtain the EMG-to-force mapping from the processed EMG and force data (calculated with the tic and toc Matlab functions), the quality of the reconstruction of a dataset, collected during a session of protocol 1 or a condition of protocol 2, with a mapping extracted from the same dataset or a dataset collected during a different session or condition, and the consistency of the mapping across sessions or conditions were tested.

The quality of the reconstruction of the end-point force was estimated through the quality of reconstruction (or fraction of data variation explained) value *R*^2^:(12)$$R^{2}=1-\frac{SS_{res}}{SS_{tot}}$$
where $SS_{res}=\underset{ij}{\sum }{\left(y_{j,i}-{\tilde{y}}_{j,i}\right)}^{2}$ was the residual sum of squares, $SS_{tot}=\underset{ij}{\sum }{\left(y_{j,i}-{\bar{y}}_{j}\right)}^{2}$ is the total sum of squares, $y_{j,i}$ the experimental data, i.e., the *j*-th end-point force component collected at the i-th time sample, ${\tilde{y}}_{j,i}$ the simulated data were calculated by multiplying the EMG-to-force matrix by the EMG signal, and ${\bar{y}}_{j}$ the j-th end-point force component averaged along the whole time interval. Therefore, an *R*^2^ value equal to 1 indicates a perfect fitting, while an *R*^2^ value equal to 0 indicates a reconstruction as good as the one obtained fitting the signal with its mean. The EMG-to-force matrix used in the calculation of the *R*^2^ of a given dataset was the one extracted from the same dataset (within-dataset fitting) or the one extracted from a dataset collected during a different session or condition (cross-dataset fitting).

The consistency measured the changes of the muscle pulling vectors, i.e., the maximum force that a muscle could exert, which is expressed by the columns of the EMG-to-force matrix, across different sessions or conditions. It was calculated as the difference between the pulling vectors of the muscles calculated during different sessions or conditions with the same approach. During protocol 1, the consistency was calculated as the difference between the pulling vectors obtained from data collected during different sessions normalized to the mean length of the two pulling vectors. This value was averaged across muscles and pairs of sessions from which the pulling vectors were extracted. During protocol 2, the consistency was calculated as the difference between the pulling vectors obtained from data collected during the baseline block with respect to the pulling vectors obtained from data collected during the perturbed blocks, normalized to the mean length of the two pulling vectors, and averaged across muscles.

An ANOVA [71,72] (Matlab function anovan) tested the effects of the following factors:Type of fitting,Session (protocol 1) or condition (protocol 2),Subject

On the computational time and the quality of the reconstruction (*R*^2^) in a within-dataset fitting.

Another ANOVA tested the fixed effects of:Type of fitting,Session from which the EMG-to-force mapping was extracted,Session from which the data were fitted (protocol 1), or whether the mapping was extracted from a perturbed block to fit the baseline block or vice-versa (protocol 2)Subject,

On the quality of the reconstruction in a cross-dataset fitting.

Finally, another ANOVA tested the effects of:Type of fitting,Subject,

On the consistency of pulling vectors.

Post-hoc analyses were performed with the paired non-parametric Wilcoxon signed rank test (Matlab function signrank) to investigate for statistical differences between different approaches or different datasets.

## 3. Results

### 3.1. Computational Time

The time required to calculate the EMG-to-force mapping with different approaches depends on the data dimensionality and on the performance of the computer used to extract the mappings. Therefore, the calculation times reported in this study depend on the specific computer, but they provide a comparison between the computational loads of the different approaches.

An ANOVA identified a significant effect of the extraction approach on computational time (Figure 3) in both protocols (*p* < 0.001).

Post-hoc tests, performed with Wilcoxon signed rank test on data collected during protocol 1 (Figure 3A), demonstrated a significant difference between all the comparisons among extraction approaches pairs (*p* < 0.002 for all comparisons). The LR was identified as the approach requiring the shortest calculation time (mean(std) across participants and session/conditions; <0.01 (0.00) s), then the MS approach (0.14 (0.05) s), and the RR approach (0.9 (1.0) s). The AD approach was identified as the most time-consuming (20 (1) s).

Post-hoc tests, performed on data collected during protocol 2 (Figure 3B), demonstrated a significant difference between all the comparisons between extraction approaches pairs (*p* < 0.004) except for the comparison between the RR and the AC approaches (*p* = 0.060). Similarly, to the results from data collected during protocol 1, the LR was identified as the approach requiring the shortest calculation time (0.04 (0.03) s), then the MS approach (1.3 (0.9) s), the RR approach (68 (121) s), and the AD approach (119 (99) s).

The extraction of the EMG-to-force mappings from data collected from a participant practicing protocol 2 took more time than the extraction from data collected from a participant practicing protocol 1 for all the extraction approaches because of the larger dimensionality of the dataset of protocol 2.

The large variability in the computational time of the RR reflected the large variability in the ridge parameter values that were calculated through an iterative approach that stopped when the ridge parameter reached a minimum of the mean squared error.

### 3.2. Within-Dataset Fitting

Figure 4 represents an example of experimental data and the fittings obtained with different approaches.

The ANOVA performed on the reconstruction quality *R*^2^ values calculated by fitting the force data as the product of the EMG-to-force mapping, extracted during the same session/condition, by the EMG data, showed a significant effect of the type of fitting (*p* < 0.001) both for data collected during protocol 1 (Figure 5A) and protocol 2 (Figure 5B). A Wilcoxon signed rank test identified a significant difference in all the comparisons between reconstruction approaches performed on data collected during both protocols (*p* ≤ 0.02 in all comparisons). As expected, the LR approach, which by definition minimizes the sum of the squared differences between experimental and reconstructed data, showed the best fitting (mean (std) of *R*^2^ across participants and sessions/conditions 0.93 (0.02) for protocol 1 and 0.81 (0.08) for protocol 2), then the RR approach showed a slightly lower, but still significantly different, fitting (*R*^2^ 0.92 (0.02) for protocol 1 and 0.79 (0.08) for protocol 2), than the AC approach (*R*^2^ 0.83 (0.18) for protocol 1 and 0.73 (0.12) for protocol 2) and the worst reconstruction was achieved with the MS approach (*R*^2^ 0.80 (0.08) for protocol 1 and 0.64 (0.15) for protocol 2). The better fitting identified for data collected during protocol 1 with respect to data collected during protocol 2 was due to the larger dimensionality of the dataset of protocol 2, which implied higher variability and, therefore, lower *R*^2^ values.

The ANOVA performed on the reconstruction *R*^2^ values calculated on data collected during protocol 1 did not identify an effect of the session (*p* = 0.93) nor of the subject (*p* = 0.09). In contrast, the ANOVA performed on the *R*^2^ values calculated during protocol 2 demonstrated an effect of the condition and the subject (*p* < 0.001 for both effects). In particular, data collected during the perturbed blocks showed a poorer fit (*p* < 0.001; *R*^2^ 0.69 (0.14)) with respect to data collected during the baseline block (*R*^2^ 0.80 (0.09)). This discrepancy was due to the larger dataset dimensionality of the perturbed blocks and to the specific co-contraction task, which implies higher levels of variability.

### 3.3. Cross-Dataset Fitting

Fitting a dataset with an EMG-to-force mapping estimated using the same dataset may lead to a good reconstruction because of the fitting of noise. A better comparison between the proposed approaches could be achieved by fitting a dataset with a mapping estimated using a different dataset. In particular, the datasets collected with protocol 1 involved the same task performed during different sessions, while protocol 2 involved the same task performed under different co-contraction conditions.

Data collected during each session of protocol 1 were fitted with the EMG-to-force mapping calculated using data collected during a different session (Figure 6A). The ANOVA performed on the reconstruction R^2^ values showed a significant effect of the type of fitting, the set of fitted data, the set of data used for estimating the mapping, and the subject (*p* < 0.03 for all effects). The paired non-parametric Wilcoxon signed rank test identified an effect of the type of estimation between LR and RR (*p* < 0.001), LR and AC (*p* = 0.020), RR and AC (*p* = 0.002), and AC and MS (*p* = 0.015), but not between LR and MS (*p* = 0.65), and RR and MS (*p* = 0.15). The best fitting was obtained with the RR approach (R^2^ 0.70 (0.24)), with the LR approach showing a slightly but significantly lower fitting (R^2^ 0.68 (0.25)), followed by the MS approach (R^2^ 0.67 (0.11)). The worst reconstruction was achieved with the AC approach (R^2^ 0.62 (0.22)).

Similarly, data collected during each condition of protocol 2 fit with the EMG-to-force mapping estimated using data collected during the other condition (Figure 6B). The ANOVA performed on the reconstruction R^2^ values showed a significant effect of the condition and the subject (*p* < 0.001 for both effects), but not of the type of fitting (*p* = 0.71). However, the mapping extracted from data from the perturbed blocks led to a better fitting [R^2^ 0.47 (0.70)] of data from the baseline block (*p* < 0.001) with respect to the fitting of data from the perturbed blocks with a mapping calculated using data from the baseline block [R^2^ −0.28 (0.78)]. Therefore, we only investigated the *R*^2^ values obtained by fitting data from the baseline block with a mapping calculated using data from the perturbed blocks. However, the approach used for the mapping calculation did not significantly affect the *R*^2^ value in any comparison (*p* > 0.19) except between AC and MS (*p* = 0.016). The best fitting was achieved with the AC approach (R^2^ 0.64 (0.12)) and then with the MS approach (R^2^ 0.54 (0.19)), then the RR approach (R^2^ 0.36 (1.00)) and the LR approach (R^2^ 0.36 (1.02)).

### 3.4. Consistency

A good cross-dataset fitting does not imply that the mappings estimated across the different sessions or conditions are consistent, i.e., are similar. In fact, an inconsistent identification of the pulling vector of a muscle that was not recruited to perform the task would still lead to a good cross-dataset fitting. Consistency of the identification of the EMG-to-force mapping across sessions or conditions could be interpreted as the ability of the algorithm to determine the same pulling force vectors. In this study, the consistency of the mapping across different sessions or conditions was assessed by comparing the pulling vectors identified for different muscles with different approaches (Figure 7).

An ANOVA comparing the differences between the pulling vectors identified with different approaches from data collected during different sessions of protocol 1 (Figure 8A), demonstrated a significant effect of subject (*p* < 0.001), but not of the pairs of compared sessions (*p* = 0.419), while there was a trend towards significance for the type of approach (*p* = 0.095). However, comparison between each pair of approaches demonstrated that the MS approach led to statistically more consistent estimations (*p* < 0.006 in comparisons with all other approaches, normalized difference: 67 (9)%), while no statistically significant differences were detected between the other approaches (*p* > 0.200, normalized difference: 84 (13)%, 83 (8)%, 89 (16)% for the LR, RR, and AC approaches, respectively).

An ANOVA comparing the differences between the pulling vectors, identified with different approaches from data collected during different conditions of protocol 2 (Figure 8B), demonstrated an effect of both subject and type of approach (*p* < 0.001 for both). Differently from results obtained from data collected from protocol 1, the comparison between each pair of approaches demonstrated that the MS approach led to statistically less consistent estimations (*p* < 0.014 with all other approaches, normalized difference: 98 (19)%), while no statistically significant difference was detected between the other approaches (*p* > 0.130, normalized difference: 78 (10)%, 76 (11)%, 74 (20)% for the LR, RR, and AC approaches, respectively).

The discrepancy between the results obtained from the two protocols could be ascribed to the nature of the differences between data sessions/conditions. In fact, while protocol 1 required the repetition of the same protocol and, therefore, the recruitment of the same set of synergies, protocol 2 likely involved the recruitment of novel co-contraction-specific synergies. Therefore, while the consistency of the mapping identified with the synergy approach during protocol 1 was guaranteed by the consistency of the muscle synergy set extracted during different sessions, the lack of consistency between the sets of muscle synergies among the two co-contraction conditions of protocol 2 did not allow to determine a consistent mapping.

## 4. Discussion

The use of EMG signals, collected from multiple muscles, to estimate the force that a human operator exerts [73] may allow to control in real-time different types of robotic devices, such as prostheses [74,75,76] or exoskeletons [77,78], and it was recently proposed as a control signal for supernumerary limbs [79]. However, some explanatory variables, which in this scenario are the muscle activations, may be linearly related, because of crosstalk [24] or physiological characteristics of the neural control of multiple muscles [63]. Therefore, a standard linear regression analysis, previously proposed in the literature, especially during isometric tasks [16,17,19,20,21,80], would be affected by multicollinearity and might lead to overfitting [81]. Since an accurate EMG-to-force mapping should estimate the end-point force according to the participant’s intention, an improper noise fitting may lead to an inaccurate reconstruction, with the consequent instability of the end-effector and inability for an operator to use myoelectric signals to control a robotic device. In this study, we compared three approaches proposed in the literature to overcome multicollinearity [35]: ridge regression [36], the implementation of foreknown anatomical constraints derived from an accurate musculoskeletal model [44], and the reduction of the number of variables to a set of physiologically meaningful combinations, i.e., the muscle synergy activation coefficients [82]. A proper definition of the EMG-to-force mapping with the different approaches was tested on data collected during two protocols, both involving the repetition of an isometric force generation task with the upper limb. While the first protocol involved the repetition of the same task across different sessions spanning two days [40], the second protocol required the generation of a set of submaximal forces in two conditions: with or without an explicit co-contraction request [41]. Figure 9 shows a summary of the results presented in this study.

The mapping provided by the LR approach, as expected, best fitted the experimental data from which it is extracted. However, this reconstruction may be driven by overfitting of noise, while fitting data through a mapping extracted from data collected during a different session/condition would allow for a better identification of the mapping associated with the voluntary physiological control, which is expected to remain unchanged across datasets, and to be less affected by noise, which depends on the datasets. Intriguingly, different conclusions may be drawn after the extraction of the mapping from the dataset collected under protocol 1 or protocol 2. While RR led to the best fitting of data collected under the same conditions during different sessions (i.e., during protocol 1), the practice of different conditions, such as those in protocol 2, requiring different co-contraction levels, led to different conclusions. First, no approaches that estimated mappings from data collected during the baseline block, i.e., without an explicit request of co-contraction modulation, were able to properly fit data collected during the perturbed blocks, despite the exerted forces being the same in the two conditions. This discrepancy could be ascribed to the specific neural control that underlies the voluntary modulation of co-contraction with respect to force generation, which might originate in a distinct anatomical portion of the premotor cortex [83] and might be driven by a separate descending pathway [34,84], with muscle synergies selectively recruited to modulate co-contraction [41]. Therefore, this analysis suggests that an EMG-to-force mapping is unable to properly fit data collected under conditions requiring the involvement of more complex control architectures, i.e., requiring novel muscle synergies or different neural pathways driving the muscles. This observation is particularly relevant for the motor rehabilitation of patients with neuromuscular diseases or during new motor skill learning. In fact, changes in the muscle patterns and muscle synergies were identified during the rehabilitation after neurological diseases [85], such as stroke [86], cerebellar ataxia [87], or multiple sclerosis [88], and during the learning of new motor skills [89,90,91,92]. These changes may lead to an incorrect fitting of data collected at later stages of the rehabilitation treatment or after a new motor skill is learned, with an EMG-to-force mapping estimated from data collected before the beginning of the treatment or during the initial stages of the practice of the new skill. In these conditions, a mapping calculated with the AC approach from data collected during the perturbed blocks would provide the best fit of data collected during the baseline block, suggesting the requirement of high variability in the data from which the EMG-to-force mapping would be extracted to achieve higher fitting accuracy. Otherwise, an inherently incremental approach [93], in which the mapping is updated online, would guarantee efficient myoelectric control even while learning novel muscle patterns.

Different conclusions could also be drawn from the investigation of the consistency of the mappings estimated during different sessions or conditions. In fact, while the muscle synergies approach determined the most consistent mapping across sessions performed under the same conditions, it also determined the less consistent mappings when considering different conditions. Since similar tasks could be achieved with the same set of synergies [94,95,96], the consistency of synergies extracted from data collected during different sessions under the same condition ensured consistency of the EMG-to-force mapping identified with the MS approach. However, more complex tasks may only be achieved by recruiting new synergies, and therefore, the inconsistency of the muscle synergy set across conditions would lead to inconsistency of the mapping identified with an MS approach.

To the best of our knowledge, this is the first study that directly compared the outcome of different algorithms, designed to overcome multi-collinearity, for the estimation of the linear relation between the activities of multiple muscles acting on a limb and the end-point force that they generate. In particular, each algorithm was designed to compensate for a specific source of the dependence between the activities of different muscles. The dependence between activities due to crosstalk [24], could be reduced by the RR algorithm. In contrast, as anatomical features bound the activities of different muscles, which may result in a correlation between these activations [97,98,99], the introduction of foreknown anatomical constraints, as in the AC algorithm, would reduce this source of dependence. Finally, as the dependence among muscle activities may be a physiological strategy to reduce the number of DoFs and lower the computational cost in the selection of motor strategies [27], constraining the EMG-to-force mapping to muscle synergies would overcome the multi-collinearity arising from the recruitment of different muscles by the same synergy. Therefore, our study not only provides a comparison between algorithms, which could be of use in the definition of the control logic of myoelectric devices, but it also indirectly suggests which source of dependence between the activities of different muscles is predominant in a dataset. A longitudinal study, as the one proposed in protocol 1, would mostly be affected by the signal crosstalk. However, as the control laws that drive muscles during different sessions are the same, i.e., the same synergy set is recruited across sessions, the MS approach, applied on data collected from different sessions, would provide consistent mappings. In contrast, the dependence between muscle activations collected under different conditions, i.e., with different impedance levels, cannot be reduced to mere crosstalk, but it also involvse different synergy sets [41] for different conditions. Therefore, the only approach, among those that we proposed allowing for the determination of an EMG-to-force mapping accurately fitting data collected under different stiffening levels is the linear fitting with anatomical constraints, which are independent of the neural control of the motor tasks. In future work, we intend to design a novel algorithm that combines all the presented approaches, providing an estimation of the EMG-to-force mapping that overcomes the multi-collinearity due to crosstalk, and anatomical, and physiological origins.

All the tested approaches showed some limitations that must be considered when selecting the most appropriate approach to estimate the EMG-to-force mapping. Although the LR represents a good solution in terms of the reconstruction and consistency when the experiment is performed under the same conditions and therefore involves the same muscle patterns, it is not the best solution when different muscle patterns are involved. Similarly, the RR approach, which is preferable to the LR because it reduces the effects of multicollinearity in longitudinal experiments requiring constant muscle patterns, and the MS approach, which leads to a consistent reconstruction across sessions, failed to fit data collected under different conditions. The best approach to be implemented in protocols with different conditions is the AC approach, involving foreknown anatomical constraints, but it requires the longest calculation time, and it may require subject-specific musculoskeletal models, e.g., in the case of amputees or neurological patients [13]. Moreover, the data optimization performed as the last step of the AC approach relies on amplitude and angle deviation boundaries, which may affect the identified mapping, and an accurate scaling of the standard Opensim model dimensions should be determined by a motion caption system, with the consequent increase of the computational time and the instrumentation costs.

This study compared different approaches through an offline analysis, but the central nervous system is able to adapt to slight changes in the body scheme [100,101,102,103,104]. Therefore, the actual selection of an approach for the EMG-to-force estimation during a myoelectrically controlled task should be determined based on how natural the participants perceive the control with a mapping extracted with the different approaches. Therefore, in future work, we plan to conduct a longitudinal study in which participants will myoelectrically control a virtual cursor with an EMG-to-force mapping extracted with the different approaches presented in this study to determine which is perceived as the most natural. Future work will also involve neurological patients whose EMG signals may show higher signal-to-noise ratios, leading to a large detrimental effect of the overfitting generated by multicollinearity. Finally, in this study, we did not consider the effect of data processing on EMG-to-force mapping, an issue that has been recently investigated [105]. Future studies will investigate whether the different approaches proposed in this study, combined with optimized data processing based on the implemented approach, may improve the fitting of the end-point force that the operator intends to exert with the EMG signal collected from multiple muscles.

## 5. Conclusions

We provided a multifaceted comparison between different approaches for the estimation of the linear mapping between multiple muscle activations, collected with surface EMG, end-point force. The comparison was performed on two datasets involving multiple sessions and different conditions. The best approach to estimate the EMG-to-force mapping is not unique and depends on the source of dependence between muscle activations expected in the specific experimental paradigm. As the different approaches selectively addressed specific sources of multi-collinearity, this study could not only be of great use in selecting the control laws for myoelectrically controlled robotic devices, but it also provides an indirect characterization of the source of dependence between the activations of different muscles. Our results suggest that, if participants are expected to recruit the same muscle patterns across the whole experiment, the RR approach or the MS approach should be preferred. Otherwise, if participants are expected to change the muscle patterns across the experiment because of different experimental conditions, the physiological effect of a motor rehabilitation therapy, or the learning of novel motor skills, the identification of the mapping should be performed on a dataset providing the largest muscle variability through the AC approach. Future work will implement the proposed approaches in controlled experiments with myoelectrically controlled devices to determine which is perceived as the most natural by participants.

## Figures and Tables

**Figure 1 bioengineering-10-00234-f001:**
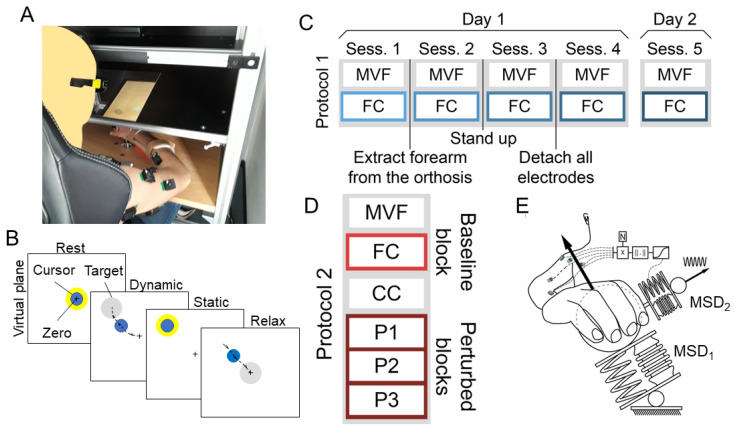
Setup and protocols. (**A**) Setup. (**B**) Task. (**C**) Protocol 1 was composed of 5 sessions: session 1 to 4 was performed during the first day, while session 5 was performed during the second day. Each session was equal and composed of a first block of Maximum Voluntary Force (MVF) and a second Force Control block (FC). Only data collected during the FC block of each session were used in this study. (**D**) Protocol 2 was composed of 6 blocks. An initial MVF block, followed by an FC block. Then a pure co-contraction block (CC) was followed by three Perturbed blocks (P1, P2, and P3). Only data collected during the baseline (FC) block and during the P1, P2, and P3 blocks, grouped together, were used in this study. (**E**) Concept of the Perturbed blocks (P1, P2, and P3) of Protocol 2, in which the displacement of the virtual cursor was simulated as two connected mass-spring-damper systems.

**Figure 2 bioengineering-10-00234-f002:**
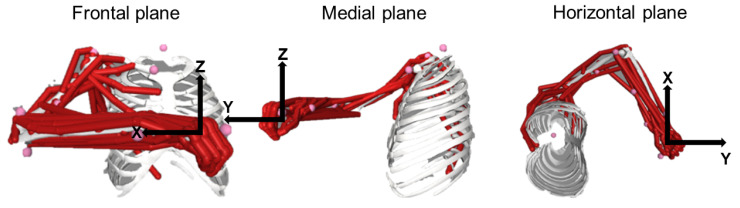
The posture at which the Musculoskeletal model was set to mimic the posture assumed by participants during the isometric task. Left panel: frontal plane; Middle panel: medial plane; Right panel: horizontal plane.

**Figure 3 bioengineering-10-00234-f003:**
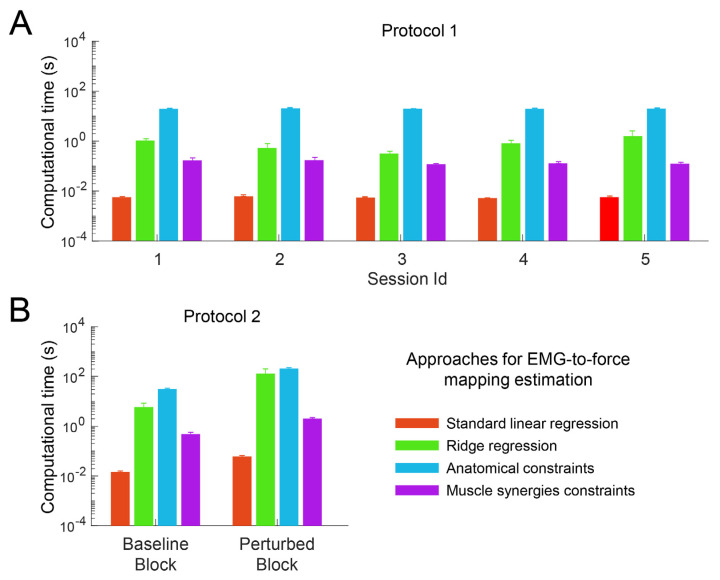
Computational time, i.e., the time required to calculate the EMG-to-force mapping with different approaches (color-coded, mean ± SE across participants). The y axis is in logarithmic scale. (**A**). results got from data collected during protocol 1. (**B**). results got from data collected during protocol 2.

**Figure 4 bioengineering-10-00234-f004:**
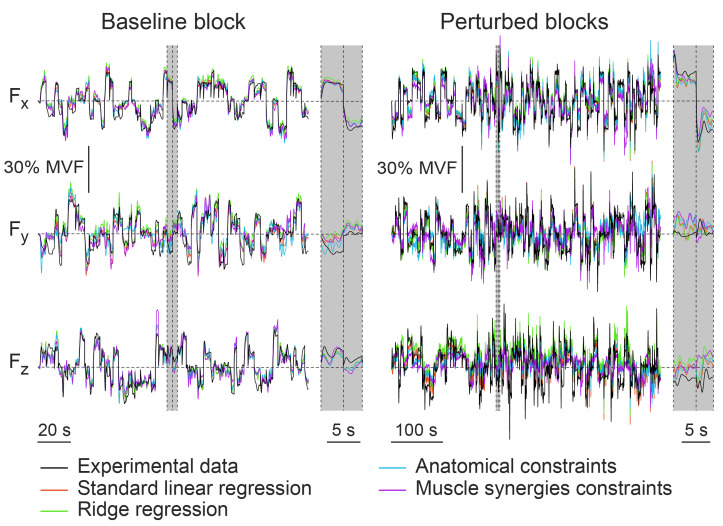
Example of the endpoint force data, fitted with EMG-to-force mapping calculated with different approaches. The experimental force data (black line), collected from a participant (#1) who practiced the protocol 2. The upper panels reported the force component collected along the x axis, the middle panels reported the force component collected along the y axis, and the lower panels reported the force component collected along the z axis. The left panels reported data collected during trials of the Baseline block, while right panels reported data collected during trials of the Perturbed blocks. Two trials of the Baseline block and of the Perturbed blocks, separated by vertical dashed lines, are highlighted in gray and magnified on the right portion of each panel.

**Figure 5 bioengineering-10-00234-f005:**
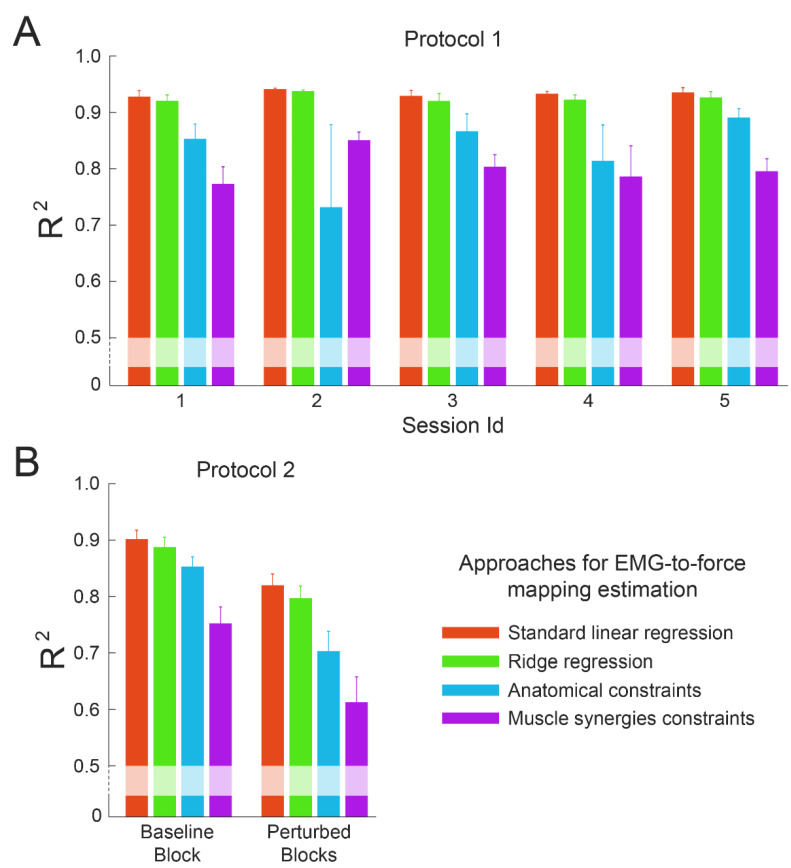
Results of the within-dataset fitting. Coefficients of determination (*R*^2^) calculated by fitting a dataset collected during protocol 1 (panel **A**) or protocol 2 (panel **B**) with an EMG-to-force mapping extracted from the same dataset. Each bar represented the coefficients of determination obtained by using different approaches to fit data (color-coded, mean ± standard error (SE) across participants).

**Figure 6 bioengineering-10-00234-f006:**
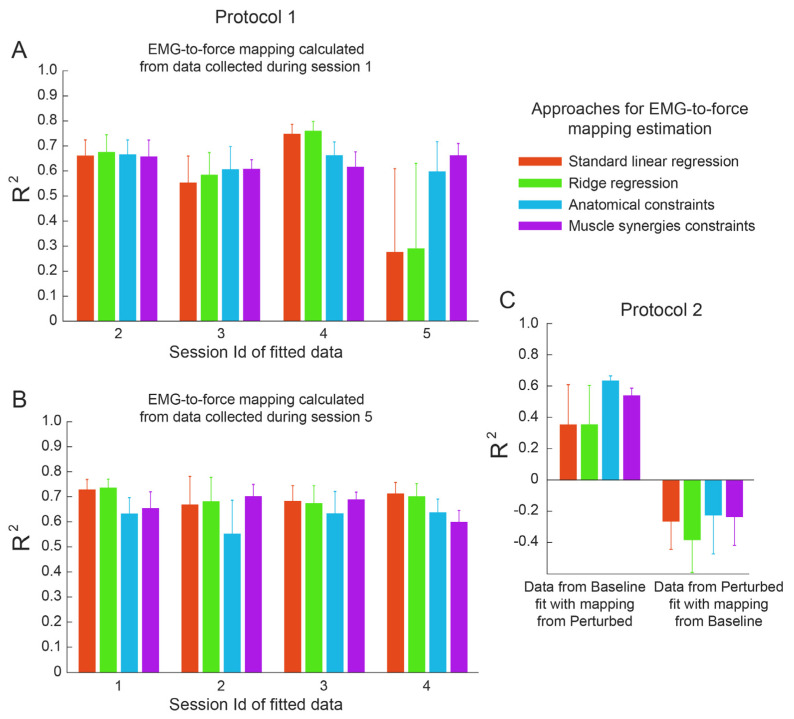
Results of the cross-dataset fitting. *R*^2^ values calculated by fitting a dataset collected during a session/condition with an EMG-to-force mapping extracted from data collected during another session/condition. Different colors are related to different estimation approaches. (**A**). *R*^2^ values (mean ± SE across participants) obtained by fitting data collected during sessions 2 to 5 of protocol 1 with the EMG-to-force mapping estimated using data collected during session 1. (**B**). *R*^2^ values obtained by fitting data collected during sessions 1 to 4 of protocol 1 with the EMG-to-force mapping estimated using data collected during session 5. (**C**). *R*^2^ values obtained by fitting data collected during the baseline block with the EMG-to-force mapping estimated using data collected during the perturbed blocks (**left**), and by fitting data collected during the perturbed blocks with the EMG-to-force mapping estimated using data collected during the baseline block (**right**).

**Figure 7 bioengineering-10-00234-f007:**
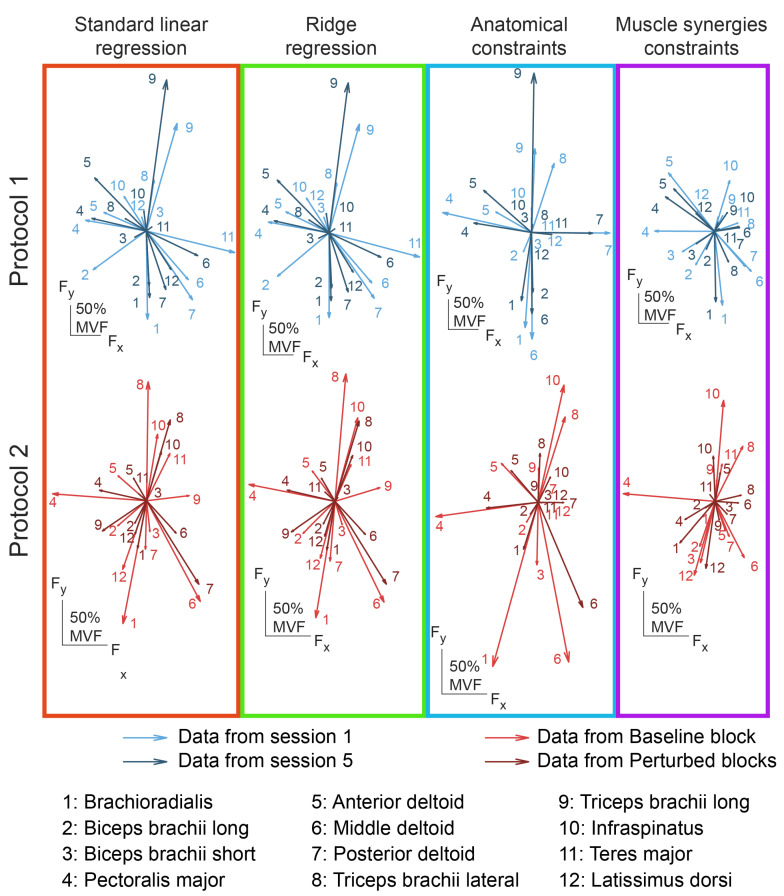
Examples of the pulling vectors of the EMG-to-force mappings calculated with different approaches. In the upper panels are reported the pulling vectors of two EMG-to-force mappings, extracted from data collected during session 1 (light blue) and session 5 (dark blue) of protocol 1, with different approaches. In the lower panels are reported the pulling vectors of two EMG-to-force mappings, extracted from data collected during baseline block (light red) and perturbed blocks (dark red) of protocol 2, with different approaches. Each column represents a different estimation approach, and each number indicates a muscle.

**Figure 8 bioengineering-10-00234-f008:**
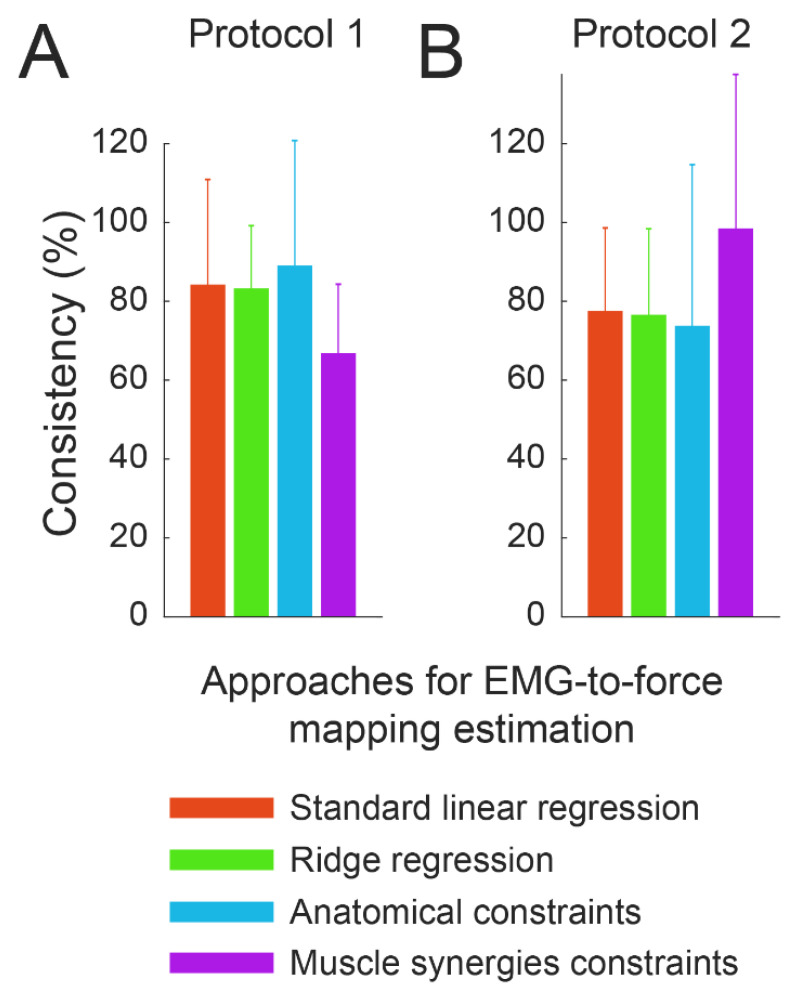
Consistency of the EMG-to-force mappings across sessions or conditions. The consistency of the pulling directions across sessions/conditions (mean ± SE across participants) was estimated as the difference between the pulling vectors of muscles, averaged across muscles and pairs of sessions for protocol 1 (panel **A**) or only across the muscles for protocol 2 (panel **B**), estimated with different approaches. Consistency is reported as a percentage of the mean between the amplitudes of the pulling vectors that were subtracted to determine it.

**Figure 9 bioengineering-10-00234-f009:**
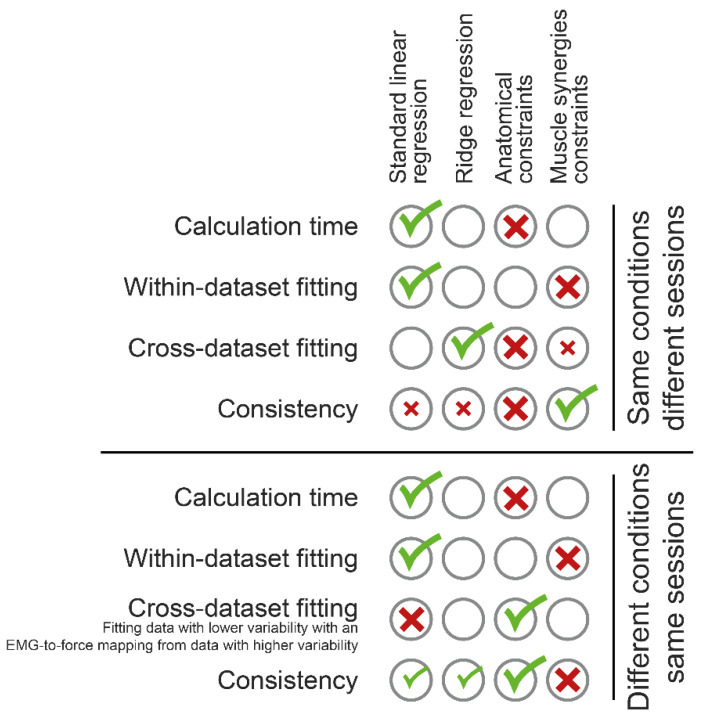
Summary of results. For each comparison metric, the best approach was indicated with a green tick, while the worst approach was indicated with a red cross. Smaller ticks indicate approaches not showing statistical difference with respect to the best approach. Similarly, smaller crosses indicate approaches not showing statistical difference with respect to the worst approach. Among the results got when different conditions were required, the only ones got after fitting a data subset with a mapping extracted from another data subset showing a higher variability, are reported in the figure.

## Data Availability

The data presented in this study are available on request from the corresponding author. The data are not publicly available due to privacy restrictions.

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
