# Peer review of "Use of Surface Electromyography to Estimate End-Point Force in Redundant Systems: Comparison between Linear Approaches"

_bioengineering, 2023, doi:10.3390/bioengineering10020234_

Round 1

Reviewer 1 Report

The authors present the paper entitled “The Estimation of the Linear Mapping Between Surface EMG and End-point Force of a Redundant System.”

This study compares the EMG-to-force mapping estimations performed with standard multiple linear regression and three other algorithms designed to reduce the detrimental effects of multi-collinearity. 

The article presents the following concerns:

  • The title could be more attractive. It seems ambiguous. Please improve it.
  • Abstract: Please provide quantitative values in order to highlight the findings.
  • The contraction for Degrees of freedom is mentioned as DOF and DoF. Please use the same term.
  • According to the aim of the manuscript. What are the main contributions of the work? It is suggested to mention the contributions in this section.
  • Line 371-382: I suggest improving the writing of these lines. It can be synthesized by listing the effects that the ANOVA will test.
  • Figure 3A: Please describe Figure 3A and Figure 3B in the main text separately.
  • Add hyperlinks to tables, figures, and references.
  • It is recommended to present a table that summarizes and compares the main findings of the proposed work vs the reported in the state-of-the-art.
  • It is recommendable to describe the structure of the text at the end of the introduction.
  • Line 372 can be supported with the following references regarding the ANOVA method: A novel method for measuring subtle alterations in pupil size in children with congenital strabismus; Statistical analysis and data envelopment analysis to improve the efficiency of the manufacturing process of electrical conductors.
  • Line 38 can be supported with the following references regarding the EMG signals: A novel methodology for classifying emg movements based on SVM and genetic algorithms; A study of computing zero crossing methods and an improved proposal for emg signals; Support vector machine-based emg signal classification techniques: a review. 
  • In general, the manuscript in its current form is easy to read. However, my biggest concern is about the level of plagiarism, about 46%. This level may compromise the innovation of the work.

The following misspelling should be checked:

  1. line 19: The article “a” may be incorrect. Consider changing it to agree with the beginning sound of the following word “L2”: “an L2”
  2. line 59: The abbreviation “i.e.” seems to be incorrectly punctuated. The correct form is “i.e.,” (the same case at the line 85)
  3. line 97: It appears that you are missing a comma after the introductoryphrase “In this study”. Add a comma. 
  4. Line 118: It appears that the phrase “by the force” is not paired with the correct article. Consider changing it by “by force”.
  5. line 170: The numeral “3” is used instead of the word spelled out. Consider spelling out the number.
  6. line 629: “In fitting” doesn´t seem to work here. Changing by “to fit”

Author Response

The authors present the paper entitled “The Estimation of the Linear Mapping Between Surface EMG and End-point Force of a Redundant System.”

This study compares the EMG-to-force mapping estimations performed with standard multiple linear regression and three other algorithms designed to reduce the detrimental effects of multi-collinearity. 

The article presents the following concerns:

We thank the reviewer for her/his comments and suggestions on our manuscript. Below is a detailed description on how we modified the manuscript according to the reported concerns.

The title could be more attractive. It seems ambiguous. Please improve it.

According to the reviewer suggestion, we changed the title form: ‘The Estimation of the Linear Mapping Between Surface EMG and End-point Force of a Redundant System’ to ‘Use of Surface Electromyography to estimate End-point Force in Redundant Systems: Comparison Between Linear Approaches’.

Abstract: Please provide quantitative values in order to highlight the findings.

The most relevant quantitative values were added in the abstract as follows:

Lines 28-32: When applied to multiple sessions, Ridge Regression achieved higher accuracy (R2 = 0.70) but estimations based on muscle synergies were more consistent (differences between the pulling vectors of mappings extracted from different sessions: 67%). In contrast, the implementation of anatomical constraints was the best solution, both in terms of consistency (R2 = 0.64) and accuracy (74%), in case of different co-contraction conditions.

The contraction for Degrees of freedom is mentioned as DOF and DoF. Please use the same term.

We thank the reviewer for spotting this typo. In the revised version of the manuscript, a unique abbreviation is used (DoF).

According to the aim of the manuscript. What are the main contributions of the work? It is suggested to mention the contributions in this section.

In the revised version, the aim of the manuscript was stated in the introduction as follows:

Lines 88-94: The aim of this study was to compare different approaches for the estimation of the EMG-to-force mapping based on linear regression and other approaches which specifically target the reduction of the effects due to different sources of multicollinearity. This comparison may be useful in the selection of the control logic of myo-electrically controlled robotic devices driven by the activities of multiple muscles, and may provide an approach to indirectly infer the main source of the dependence between the activities of different muscles in a dataset.

Line 371-382: I suggest improving the writing of these lines. It can be synthesized by listing the effects that the ANOVA will test.

The effects that were tested with the different ANOVA analyses were listed to improve the writing (Lines 382-400).

Figure 3A: Please describe Figure 3A and Figure 3B in the main text separately.

We thank the reviewer for her/his suggestion, that we agree would make the presented results clearer. Results described in Figures 3A and 3B are separated in two paragraphs in the new revision (Lines 418-424 describes Figure 3A; Lines 425-430 describes Figure 3B).

Add hyperlinks to tables, figures, and references.

We agree that hyperlinks would provide the reader with an easy access to figure, tables, and references, but we are afraid that this may lead to some issues in the post-acceptance formatting process. So, we preferred not to add them.

It is recommended to present a table that summarizes and compares the main findings of the proposed work vs the reported in the state-of-the-art.

The main findings are summarized in Figure 9, which provides a quick overview on the best and worst algorithm according to the type of data and aspect which was investigated. Unfortunately, literature does not provide similar investigations, which is the reason for which this study was required.

It is recommendable to describe the structure of the text at the end of the introduction.

At the end of the introduction, the structure of the text was described adding the following paragraph:

Lines 108-112: The present work is organized as follows. In Section 2, the experimental setups and protocols are presented, and the algorithms for the identification of the EMG-to-force mappings and the analyses implemented to compare them are described. The findings are reported in Section 3. Finally, the Discussion and Conclusions are found in Sections 4 and 5, respectively.

Line 372 can be supported with the following references regarding the ANOVA method: A novel method for measuring subtle alterations in pupil size in children with congenital strabismus; Statistical analysis and data envelopment analysis to improve the efficiency of the manufacturing process of electrical conductors.

The suggested references were added in the new version of the manuscript.

Line 38 can be supported with the following references regarding the EMG signals: A novel methodology for classifying emg movements based on SVM and genetic algorithms; A study of computing zero crossing methods and an improved proposal for emg signals; Support vector machine-based emg signal classification techniques: a review. 

We thank the reviewer for suggesting us these interesting studies, which were added in the new version of the manuscript

In general, the manuscript in its current form is easy to read. However, my biggest concern is about the level of plagiarism, about 46%. This level may compromise the innovation of the work.

 The plagiarism level, detected by the software provided by the journal editor, was 33%, where 11% was from our previous conference paper (‘Unconstrained and constrained estimation of a linear EMG-to-force mapping during isometric force generation’, doi: 10.1109/MeMeA54994.2022.9856461) in which we introduced the anatomically constrained approach (AC), and 9% was from our previous recently published article (Virtual Stiffness: A Novel Biomechanical Approach to Estimate Limb Stiffness of a Multi-Muscle and Multi-Joint System, doi: 10.3390/s23020673) in which we presented the same setup used for protocol 2 in this study. The remaining 13% was detected on 59 different sources. The methods sections were rewritten to better fit the content of this paper and to avoid misleading plagiarism detection.

The following misspelling should be checked:

line 19: The article “a” may be incorrect. Consider changing it to agree with the beginning sound of the following word “L2”: “an L2”

line 59: The abbreviation “i.e.” seems to be incorrectly punctuated. The correct form is “i.e.,” (the same case at the line 85)

line 97: It appears that you are missing a comma after the introductory phrase “In this study”. Add a comma. 

Line 118: It appears that the phrase “by the force” is not paired with the correct article. Consider changing it by “by force”.

line 170: The numeral “3” is used instead of the word spelled out. Consider spelling out the number.

line 629: “In fitting” doesn´t seem to work here. Changing by “to fit”

We thank the reviewer for spotting these typos, that we corrected in the revised version of the manuscript.

Reviewer 2 Report

A very interesting paper.

The study focuses on prediction of the end-point of a working device (either hand or robotic manipulator) with help of EMG (surface EMG). Such approach looks very promising and practical. Besides technical aspects, such approach would have helped understanding physiology of motion in real living objects in terms of "muscle synergies", "motor commands", etc. In other words, this manuscript has strong physiological aspect. I haven't got major comments.

 Minor comment: Line 235 - Should it be "regularization" instead of "regolarization?"

Author Response

A very interesting paper.

The study focuses on prediction of the end-point of a working device (either hand or robotic manipulator) with help of EMG (surface EMG). Such approach looks very promising and practical. Besides technical aspects, such approach would have helped understanding physiology of motion in real living objects in terms of "muscle synergies", "motor commands", etc. In other words, this manuscript has strong physiological aspect. I haven't got major comments.

We thank the reviewer for her/his positive feedback on our manuscript.

 Minor comment: Line 235 - Should it be "regularization" instead of "regolarization?"

We thank the reviewer for spotting this typo, that we corrected in the revised version of the manuscript.

Reviewer 3 Report

The manuscript was prepared very well. The introduction section justifies the purpose of the study. I congratulate the authors for the preparation of the manuscript

However, I have the following comments:

Title

·       avoid acronyms like EMG

Introduction

·       indicate some characteristics, types, and uses of electromyography (you can help from: doi.org/10.3390/su12219137)

·       Indicate why this study is necessary

Methods

·       It does not require any modification. They are perfectly explained and designed.

Results /

·       It is one of the strong parts of the manuscript.

·       Figures must improve their footnote for the reader to improve their understanding

Discussion

·       Include a section on strengths.

·       What does this article contribute to, the authors should make their own assessment and include their own discussion of the results shown in the manuscript?

 Conclusion

·       state the most important outcome of your work. Do not simply summarize the points already made in the body — instead, interpret your findings at a higher level of abstraction. Show whether, or to what extent, you have succeeded in addressing the need stated in the Introduction (or objectives).

Author Response

The manuscript was prepared very well. The introduction section justifies the purpose of the study. I congratulate the authors for the preparation of the manuscript

We thank the reviewer for her/his positive feedback on our manuscript.

However, I have the following comments:

Below is a detailed description on how we modified the manuscript according to the reported comments.

Title

  • avoid acronyms like EMG

The acronym in the title was expanded

Introduction

  • indicate some characteristics, types, and uses of electromyography (you can help from: doi.org/10.3390/su12219137)

A brief introduction on the EMG characteristics, types and uses were added at the beginning of the Introduction. The suggested review was considered and cited. The new text is following:

Lines 39-46: A measure of the muscle activation is provided by electromyography (EMG), i.e., the recordings of electrical in muscle fibers driven by motoneurons. While the EMG signal recorded through needle electrodes provides an accurate measure of a small volume of the muscle, applications which require the modulation of the whole muscle commonly require the use of non-invasive surface EMG. Surface EMG has been implemented in industrial applications, or research studies on motor control, confining the use of needle EMG to clinical applications, or to research investigating the recruitment of single motor neurons [1].

  • Indicate why this study is necessary

In the revised version, the purpose of the manuscript was stated in the introduction as follows:

Lines 88-94: The aim of this study was to compare different approaches for the estimation of the EMG-to-force mapping based on linear regression and other approaches which specifically target the reduction of the effects due to different sources of multicollinearity. This comparison may be useful in the selection of the control logic of myo-electrically controlled robotic devices driven by the activities of multiple muscles, and may provide an approach to indirectly infer the main source of the dependence between the activities of different muscles in a dataset.

Methods

  • It does not require any modification. They are perfectly explained and designed.

We thank the reviewer for the positive opinion given on this section of the manuscript

Results /

  • It is one of the strong parts of the manuscript.
  • Figures must improve their footnote for the reader to improve their understanding

The footnote of figures in the results section are revised to improve their readability. At the beginning of each footnote, a short description of what is reported in the figure was added.

Discussion

  • Include a section on strengths.
  • What does this article contribute to, the authors should make their own assessment and include their own discussion of the results shown in the manuscript?

In the new version we merged these two points in a single new paragraph, in which we enlightened the novelty of the approach and we added new interpretations of the results that we obtained, which may suggest which is the main source of multi-collinearity in the dataset. Following is the new paragraph:

Lines 654-683: By the best of our knowledge, this is the first study which directly compared the outcome of different algorithms, designed to overcome multi-collinearity, for the estimation of the linear relation between the activities of multiple muscles acting on a limb and the end-point force that they generate. In particular, each algorithm was designed to compensate a specific source of the dependence between the activities of different muscles. The dependence between activities due to crosstalk [24], could be reduced by the RR algorithm. In contrast, as anatomical features bounds the activities of different muscles, which may result in a correlation between these activations [97–99], the introduction of foreknown anatomical constraints, as in the AC algorithm, would reduce this source of dependence. Finally, as the dependence among muscle activities may be a physiological strategy to reduce the number of DoFs and lower the computational cost in the selection of motor strategies [27], constraining the EMG-to-force mapping to muscle synergies would overcome the multi-collinearity which arise after the recruitment of different muscles by the same synergy. Therefore, our study, does not only provide a comparison between algorithms, which could be of use in the definition of the control logic of myo-electric devices, but it may also indirectly suggest which source of dependence between the activities of different muscles is predominant in a dataset. A longitudinal study, as the one proposed in protocol 1, would mostly be affected by the signal crosstalk. However, as control laws which drive muscles during different sessions are the same, i.e., the same synergy set is recruited across sessions, the MS approach, applied on data collected from different sessions, would provide consistent mappings. In contrast, the dependence between muscle activations collected under different conditions, i.e., with different impedance levels, cannot be reduced to a mere crosstalk, but also involve different synergy sets [41] for different conditions. Therefore, the only approach, among those that we proposed, which allowed to determine an EMG-to-force mapping accurately fitting data collected under different stiffening levels, is the linear fitting with anatomical constraints, which are independent from the neural control of the motor tasks. In future works, we intend to design a novel algorithm that combines all the presented approaches, providing an estimation of the EMG-to-force mapping that overcomes the multi-collinearity due to crosstalk, and anatomical, and physiological origins.

 Conclusion

  • state the most important outcome of your work. Do not simply summarize the points already made in the body — instead, interpret your findings at a higher level of abstraction. Show whether, or to what extent, you have succeeded in addressing the need stated in the Introduction (or objectives).

We stated in the conclusion the most important outcome of our study, which is not the only comparison between the approaches for the practical purpose of define the control law of robotic devices, but, as suggested by the reviewer, also the physiological interpretations of our results. Following is the new paragraph:

Lines 718-727: In conclusion, we provided a multifaceted comparison between different approaches for the estimation of the linear mapping between end-point force and multiple muscle activations, collected with surface EMG. The comparison was performed on two datasets involving multiple sessions and different conditions. The best approach to estimate the EMG-to-force mapping is not unique and depends on the source of dependence between muscle activations expected in the specific experimental paradigm. As the different approaches proposed, selectively addressed specific sources of multi-collinearity, this study could not only be of great use in selecting the control laws for myo-electrically controlled robotic devices, but it also provides an indirect characterization of the source of dependence between the activations of different muscles.

Round 2

Reviewer 1 Report

The manuscript can be acepted for publication